# Identification of GOLPH3 Partners in *Drosophila* Unveils Potential Novel Roles in Tumorigenesis and Neural Disorders

**DOI:** 10.3390/cells10092336

**Published:** 2021-09-06

**Authors:** Stefano Sechi, Angela Karimpour-Ghahnavieh, Anna Frappaolo, Laura Di Francesco, Roberto Piergentili, Eugenia Schininà, Pier Paolo D’Avino, Maria Grazia Giansanti

**Affiliations:** 1Istituto di Biologia e Patologia Molecolari del CNR, c/o Dipartimento di Biologia e Biotecnologie, Sapienza Università di Roma, Piazzale A. Moro 5, 00185 Roma, Italy; stefano.sechi@cnr.it (S.S.); angela.karimpourghahnavieh@uniroma1.it (A.K.-G.); anna.frappaolo@cnr.it (A.F.); roberto.piergentili@uniroma1.it (R.P.); 2Dipartimento di Scienze Biochimiche A. Rossi Fanelli, Sapienza Università di Roma, Piazzale A. Moro 5, 00185 Roma, Italy; lau.difra@gmail.com (L.D.F.); eugenia.schinina@uniroma1.it (E.S.); 3Department of Pathology, University of Cambridge, Tennis Court Road, Cambridge CB2 1QP, UK; ppd21@cam.ac.uk

**Keywords:** *Drosophila*, cell cycle, GOLPH3, Golgi, FMRP, male meiosis, spermatogenesis

## Abstract

Golgi phosphoprotein 3 (GOLPH3) is a highly conserved peripheral membrane protein localized to the Golgi apparatus and the cytosol. GOLPH3 binding to Golgi membranes depends on phosphatidylinositol 4-phosphate [PI(4)P] and regulates Golgi architecture and vesicle trafficking. GOLPH3 overexpression has been correlated with poor prognosis in several cancers, but the molecular mechanisms that link GOLPH3 to malignant transformation are poorly understood. We recently showed that PI(4)P-GOLPH3 couples membrane trafficking with contractile ring assembly during cytokinesis in dividing *Drosophila* spermatocytes. Here, we use affinity purification coupled with mass spectrometry (AP-MS) to identify the protein-protein interaction network (interactome) of *Drosophila* GOLPH3 in testes. Analysis of the GOLPH3 interactome revealed enrichment for proteins involved in vesicle-mediated trafficking, cell proliferation and cytoskeleton dynamics. In particular, we found that dGOLPH3 interacts with the *Drosophila* orthologs of Fragile X mental retardation protein and Ataxin-2, suggesting a potential role in the pathophysiology of disorders of the nervous system. Our findings suggest novel molecular targets associated with GOLPH3 that might be relevant for therapeutic intervention in cancers and other human diseases.

## 1. Introduction

Golgi phosphoprotein 3 (GOLPH3) is a highly conserved phosphatidylinositol 4-phosphate [PI(4)P] binding protein, required for maintenance of Golgi structures and protein trafficking [1]. GOLPH3 function has been also involved in multiple vesicular routes including vesicular transport to the plasma membrane and intra-Golgi and endocytic trafficking [1,2,3,4,5,6]. Many studies revealed that GOLPH3 is required for coatomer (COPI)-mediated Golgi trafficking of several protein glycosyltransferases [7,8,9,10,11,12,13]. All Golgi glycosyltransferases are type II membrane proteins containing a small cytosolic N-terminal region, a single transmembrane domain and a luminal enzymatic domain [14]. Vps74p, the yeast homolog of GOLPH3, binds to the COPI coatomer as well as to a (F/L)(L/I/V)XX(R/K) motif, contained in the N-terminal tail of most yeast Golgi-resident glycosyltransferases, thus contributing to the intra-Golgi localization of these enzymes [11,12]. Similarly, human GOLPH3 binds to and recruits a class of Golgi glycosyltransferases, including Core 2 N-acetylglucosaminyltransferase 1 [15], α2,6-sialyltransferase I [7] and several glycoenzymes in the glycosphingolipid pathway [10]. *Drosophila* GOLPH3 (dGOLPH3) is required for retaining to the Golgi exostosins, a class of glycosyltransferases implicated in O-glycosylation of heparan sulfate proteoglycans [16]. 

Human GOLPH3 protein is also involved in endocytic trafficking through the retromer, the endosomal complex that regulates trafficking between the endosomes and *trans*-Golgi network (TGN) and endosome-to-plasma membrane transport [17]. Additionally, our previous study indicated the role of dGOLPH3 in early endocytic trafficking [4]. 

GOLPH3 is an oncogene that is frequently amplified in several human solid tumors including melanoma, breast cancer, glioma, lung, and colorectal cancer [5,17]. Many research studies have shown that the overexpression of GOLPH3 correlates with tumor metastasis and poor prognosis in several cancer types, including breast cancer [18] and glioblastoma [19,20]. The oncogenic activity of GOLPH3 has been associated with its ability to enhance growth factor-induced mammalian target of rapamycin (mTOR) signaling [17]. Although the connection with the mTOR pathway needs further investigation, the roles of GOLPH3 in protein trafficking and glycosylation suggest that it might contribute to cellular transformation by affecting the internalization and recycling of key signaling molecules and/or glycosylation of cancer-relevant glycoproteins/glycolipids [5,10]. We recently demonstrated that GOLPH3 is required for contractile ring assembly during cytokinesis in *Drosophila* [4,21,22]. dGOLPH3 accumulates at the cleavage site of both dividing spermatocytes and neuroblasts and interacts with contractile ring proteins and vesicle trafficking proteins. We showed that the function of dGOLPH3 in cytokinesis is intimately connected to its ability to bind PI(4)P, suggesting that it might coordinate PI(4)P signaling and membrane trafficking with contractile ring dynamics. *Drosophila* spermatogenesis provides an ideal model system to investigate the role of vesicle trafficking proteins during male meiotic division and the cytoskeleton-based morphological changes that characterize germ cell differentiation [23,24,25,26,27]. Here, we have characterized the GOLPH3 interactome in *Drosophila* testes in order to elucidate the molecular mechanisms underpinning GOLPH3 functions. Our findings could aid in the identification of novel molecular targets for therapeutic intervention of human diseases characterized by the deregulation of GOLPH3.

## 2. Materials and Methods

### 2.1. Fly Stocks and Transgenes

Flies were reared according to standard procedures at 25 °C unless otherwise noted. Oregon-R flies were used as wild-type controls unless otherwise specified. The following fly stocks were from Bloomington *Drosophila* Stock Center (Indiana University): *UASp-YFP-Rab7* (#23641, [28]); *UASp-YFP-Rab8* (#9782, [28]) *UASp-YFP-Rab10* (#9789, [28]); *UASp-γCOP-mRFP* (#29714), *UAS-δCOP.HA* (#55059), *GFP-Rac1* (#52285), *GFP-Rac2* (#52287). The line *Sec22^fTRG^* carrying a fosmid construct, expressing the specific GFP fusion protein at endogenous levels, was obtained from Vienna *Drosophila* Resource Center, Vienna Biocenter (# 318332, [29]). The *UAS-Atx2-3xHA* line was obtained from FlyORF, University of Zurich (#F001031, [30]). The *bam-GAL4* line [31] was used as a driver to express *YFP-Rab8*, *YFP-Rab10*, *YFP-Rab7*, *δCOP-HA* and *Atx2-3xHA* in spermatocytes from the *UAS* constructs. The mRFP line, used as a control, was obtained by cloning the mRFP sequence into pCasper4-tubulin and was described in [32]. Flies expressing GFP–Cog7 were previously described [33]. 

### 2.2. Molecular Cloning

*dGOLPH3-mRFP* was generated by cloning full-length *Drosophila GOLPH3* (dGOLPH3, *sauron*, *sau*) cDNA into a pCasper4-tubulin [4] in frame with C-terminal mRFP. *dGOLPH3-mRFP* was crossed into the *dGOLPH3* (*sau*^z*2217*^*/Df(2L)Exel7010*) mutant background to test for phenotypic rescue of male sterility and meiotic cytokinesis failure. To generate the RFP-βCOP fusion construct, the cDNA of βCOP was cloned into a pCasper4-tubulin in frame with N-terminal mRFP. The GFP cDNA was cloned into a pCasper4-tubulin to generate the GFP transgenic flies. Transgenic flies were generated by P-element mediated germline transformation, performed by Bestgene Inc. (Chino Hills, CA, USA).

### 2.3. Co-Immunoprecipitation Experiments

Co-immunoprecipitation (Co-IP) experiments were performed as described in [27]. For the experiments shown in Figures 3 and 4, 400 adult testes of each genotype were homogenized in 500 µL of lysis buffer [25 mM Tris-HCl (pH 7.4), 150 mM NaCl, 1mM EDTA, 1% NP-40] with protease inhibitors (#11697498001, Roche, Basel, Switzerland) on ice using a Dounce homogenizer. For the experiments of AP-MS, 3000 adult testes from either dGOLPH3-RFP or RFP males, were homogenized on ice in 1ml of lysis buffer with protease inhibitors using a Dounce homogenizer. Lysates were clarified by centrifugation and protein concentration was quantified using a NanoDrop 2000c Spectrophotometer (Thermo Fisher Scientific, Waltham, MA, USA). 4% of each lysate was retained as the “input”. The remainder was precleared with control agarose beads (bab-20, ChromoTek, Planegg, Germany). Co-IP experiments from lysates expressing GFP or RFP-tagged proteins, were performed using GFP/RFP trap-A purchased from ChromoTek (#gta-100, #rta-20), following the protocol previously described [33]. The beads were rinsed once with ice-cold IP Lysis buffer and washed extensively (4 × 5 min) on the wheel at 4 °C. After the final wash, the beads were resuspended in 30 µL of SDS sample buffer [20% glycerol, 4% SDS, 0.2% BBF, 100 mM Tris-HCl (pH 6.8), 200 mM DTT] and boiled for 10 min. 

To immunoprecipitate dGOLPH3, the testis extract from 400 adult testes expressing dAtx2-HA was precleared with Protein A-Agarose (SC-2001, Santa Cruz, Biotechnology, Dallas, TX, USA) and divided into two. Fractions were incubated with either 4 µg of rabbit anti-GOLPH3 antibody L11047/G49139/77 [4] or 4 µg of rabbit pre-immune serum L11047/G49139 [4] from the same animal before the immunization. After antibody incubation, Co-IP was carried out using the Protein A-Agarose (#SC-2001, Santa Cruz) following the manufacturer’s instructions. Co-IP experiments were performed in triplicate with identical results.

### 2.4. Western Blotting 

Samples were separated on Mini-protean TGX precast gels (Bio-Rad Laboratories, Hercules, CA, USA) and blotted to PVDF membranes (Bio-Rad). Membranes were blocked in Tris-buffered saline (Sigma-Aldrich, St. Louis, MO, USA) with 0.05% Tween-20 (TBST), containing 5% non-fat dry milk (Bio-Rad; Blotting Grade Blocker) for 1h at room temperature followed by incubation with primary and secondary antibodies diluted in TBST. Primary antibodies used for immunoblotting were as follows: mouse anti-dynamin (1:1000; Clone 41, #610246, BD Biosciences, San Jose, CA, USA), mouse monoclonal anti-dFmr1 (1:5000, clone 6A15, Sigma-Aldrich, #F4554), rabbit anti-GFP (1:2500; TP-401, Torrey Pines Biolabs, Secaucus, NJ, USA), mouse monoclonal anti-hemagglutinin (HA) tag (1:1000, clone 12CA5, #11583816001, Roche), mouse monoclonal anti-RFP (1:1000; # 6G6, Chromotek), rabbit anti-dGOLPH3 ([4], 1:2500; #G49139/77) guinea pig anti-αCOP (1:2000, [34]), rabbit anti-Sec31 (1:5000, [35]), rabbit anti-SH3PX1 (1:2000, [36]). HRP-conjugated secondary antibodies were as follows: goat anti-mouse IgG (H+L) (#31431, Pierce Biotechnology Inc., Waltham, MA, USA), goat anti-rabbit IgG (H+L) (#31466, Pierce), goat anti guinea-pig IgG (#AP108P, Sigma-Aldrich). All secondary antibodies were used at 1:5000. After incubation with the antibodies, blots were washed (3 × 5 min) in TBS-T (20 mM Tris-HCl pH 7.5, 150 mM NaCl, 0.05% Tween 20). Blots were imaged using ECL (#XLS142, Cyanagen, Bologna, Italy) and signals revealed with the ChemiDoc XRS imager (BioRad). 

### 2.5. Glutathione S-Transferase (GST) Pull-Down Assays

GST and GST-dGOLPH3 proteins were expressed in bacteria and purified using glutathione–Sepharose 4B beads (#17-0756-01, GE Healthcare, Arlington Heights, IL, USA) following the manufacturer’s instructions, as described previously [27,37]. GST pull-down experiments were performed with testis lysates using the procedure described in [38]. Testis lysates were incubated with either GST or GST–dGOLPH3 (at the appropriate concentration), bound to glutathione–Sepharose 4B beads, with gentle rotation at 4 °C for 2 h. After rinsing in wash buffer (25 mM Tris-HCl pH 7.4, 150 mM NaCl, 1% NP-40, 1 mM EDTA, Protease and phosphatase inhibitors) three times, the beads were boiled in SDS sample buffer and separated by SDS-PAGE. Bound proteins were analyzed by Western Blotting. Before immunoblotting, PVDF membranes were stained with Ponceau (#P3504, Sigma-Aldrich). GST pull-down experiments were performed in triplicate with identical results.

### 2.6. Proteomics and Data Analysis

Visualization of protein bands was obtained using a colloidal Coomassie staining. From each SDS-PAGE lane, ten slices were excised and submitted to a trypsin proteolysis [39]. Peptide mixtures were then extracted from the gel matrix and submitted to a desalting step by solid phase extraction before mass spectrometric analyses [40]. Nano-liquid chromatography tandem mass spectrometry (nanoLC-MS/MS) analyses were performed using an Ultimate3000 system (Thermo Fisher Scientific) equipped with a splitting cartridge for nanoflows and connected on-line via a nanoelectrospray ion source (Thermo-Fisher Scientific) to an LTQ-Orbitrap XL mass spectrometer (Thermo-Fisher Scientific). Each sample was automatically loaded from the autosampler module of the Ultimate 3000 system at a flow rate of 20 µL/min onto a trap column (AcclaimPepMap µ-Precolumn, 300 µm × 1 mm, Thermo Fisher Scientific) in 4% ACN containing 0.1% FA. After 4 min, peptides were eluted at 300 nL/min onto a 15 cm column (360 µm OD × 75 μm ID, 15 µm Tip ID; PicoFrit, New Objective, Littleton, MA, USA) and custom packed by reverse phase (C18.5 µm particle size, 200 Å pore size; Magic C18AQ, Michrom Bioresources, Auburn, CA, USA) using a two-step gradient of solvent B (from 5% to 40% in 120 min and from 40% to 85% in 15 min). Data-dependent tandem mass spectroscopy (MS/MS) was performed using full precursor ion scans (MS1) collected at 30,000 resolution, with an automatic gain control (AGC) of 1 × 106 ions and a maximal injection time of 1000 ms. The 5 most intense (>200 counts) ions with charge states of at least +2 were selected for collision-induced dissociation (CID). Dynamic exclusion was active, with 90 ms exclusion for ions selected twice within a 30 ms window. For MS/MS scanning, the minimum MS signal was set to 500, activation time to 30 ms, target value to 10,000 ions and injection time to 100 ms. All MS/MS spectra were collected using a normalized collision energy of 35% and an isolation window of 2 Th. All MS/MS samples were analyzed using the software package MaxQuant (version 1.3.0.5, Max Planck Institute of Biochemistry, Martinsried, Germany). Peptides sequences were searched against the *Drosophila melanogaster* Uniprot proteome database and common contaminant proteins. 

We set oxidation (methionine) and phosphorylation (serine, tyrosine, threonine) as variable modifications, carbamidomethylation (cysteine) as a fixed modification, mass tolerance of 20 ppm for the precursor ion (MS) and of 0.5 Da for the fragment ions (MS/MS). High-confidence peptide-spectral matches were filtered at <1% false discovery rate. Proteins recognized as having a low confidence level [i.e., (i) number of unique peptides ≤ 0, (ii) identified only by a modified peptide, (iii) less than 3 MS/MS spectra] were filtered out. Individual MS/MS spectra were manually inspected for proteins represented by a single tryptic peptide.

### 2.7. Computational Analysis of the dGOLPH3 Interactome

Protein classes and GO over-representation analyses were performed using the PANTHER database [41], while GO enrichment analysis was performed using the GOrilla tool [42]. Prism 9 (GraphPad Software, San Diego, CA, USA) and Excel (Microsoft Corporation, Redmond, WA, USA) software were used for statistical analyses and to prepare graphs.

### 2.8. Immunofluorescence Analysis and Live Imaging of Testes

Cytological preparations were made using testes from third instar larvae. Images of living spermatocytes expressing dGOLPH3-mRFP and GFP-Cog7 were captured as described in [32]. For immunofluorescence analysis, larval testes were fixed in 4% methanol-free formaldehyde (Polysciences, Warrington, PA, USA), squashed under a coverslip and frozen in liquid nitrogen. After removal of the coverslip, the samples were rinsed in PBS and blocked for 20 min in PBS containing 0.1% Tween-20 and 3% BSA before immunostaining. The primary antibodies included: mouse monoclonal anti-dFmr1 (1:600, clone 6A15, #F4554, Sigma-Aldrich), rabbit anti-HA (1:600, clone C29F4, #3724, Cell Signaling Technology, Danvers, MA, USA) and rabbit anti-dGOLPH3 (1:1500, [4]). The secondary antibodies included: Alexa 555-conjugated anti-rabbit IgG (1:300, #A21430, Life Technology, Carlsbad, CA, USA) and Alexa Fluor-488 anti-mouse IgG (1:400, #115-546-006, Jackson ImmunoResearch Laboratories, West Grove, PA, USA). All of the incubations with the primary antibodies (diluted in PBS containing 0.1% Tween-20 and 3% BSA) were performed overnight at 4 °C. Incubations with the secondary antibodies were performed at room temperature for 1 h. After immunostaining, samples were mounted in Vectashield Vibrance Antifade Mounting Medium containing DAPI (#H-1800, Vector Laboratories, Burlingame, CA, USA). Images were captured with a charged-coupled device (CCD camera, Qimaging QICAM Mono Fast 1394 Cooled) connected to a Nikon Axioplan epifluorescence microscope (Nikon, Minato, Tokyo, Japan) equipped with an HBO 100-W mercury lamp and a 100× objective.

### 2.9. Proximity Ligation Assay 

Larval testes were dissected in PBS and fixed using 4% methanol-free formaldehyde in PBS. Samples were blocked with the blocking solution contained in the kit (Duolink In Situ PLA Probes, #DUO92001/DUO92005, Sigma-Aldrich), following the instructions provided by the manufacturer. After blocking, samples were incubated with primary antibodies diluted in Duolink In Situ Antibody Diluent included in the kit (Duolink In Situ PLA Probes, Sigma-Aldrich) overnight in a humid chamber at 4 °C. Monoclonal antibodies were used to stain dFmr1 (1:600, clone 6A15, #F4554, Sigma-Aldrich). Polyclonal antibodies were: anti-HA (1:600, clone C29F4, # 3724, Cell Signaling) mouse anti-Rab1 (1:750, antibody S12085a [27]) and rabbit anti-dGOLPH3 (1:1500, [4]). The PLA probe incubation and the detection protocol were performed in accordance with the procedures described in the Duolink In Situ-Fluorescence User Guide, using the Duolink In Situ PLA Probes and Duolink In Situ Detection Reagents (#DUO92013/DUO92014, Sigma-Aldrich). Following the detection steps, specimens were mounted in Vectashield Vibrance Antifade Mounting Medium containing DAPI (#H-1800, Vector Laboratories). Images were captured with a charged-coupled device (Axiocam 503, mono CCD camera) connected to a Zeiss Cell Observer Z1 microscope (Carl Zeiss AG, Oberkochen, Germany) equipped with an HXP 120 V inclusive built-in power supply, lamp module and a 63X/1.4 objective. Images were acquired using the ZEN2 software along the *z*-axis. Projections were created using the Extended Depth of Focus function of the ZEN2 software and processed in Photoshop. Quantification of the number of PLA signals per cell was obtained using the Analyze Particles tools of the ImageJ software. Number of PLA signals, compared to background signals in the control, was examined for statistical significance using the nonparametric Mann-Whitney test.

## 3. Results

### 3.1. Identification of the dGOLPH3 Interactome in Drosophila melanogaster

To identify the in vivo interactome of dGOLPH3, we performed affinity purification from testis extract of flies expressing either dGOLPH3-RFP or RFP alone as control. We first assessed whether dGOLPH3-RFP localized to the Golgi in testes co-expressing dGOLPH3-RFP and GFP-Cog7 by time-lapse imaging. dGOLPH3-RFP co-localized with GFP-Cog7 [33] to the Golgi organelles of primary spermatocytes and to the ribbon-like acroblast of spermatids (Figure 1).

dGOLPH3-RFP and its associated partners were characterized by RFP affinity purification, coupled with mass spectrometry (AP-MS), and selected interactors were subsequently validated by co-immunoprecipitation (Co-IP) or gluthatione S-transferase (GST) pull-down. 

Using the MaxQuant searching platform, we identified a list of dGOLPH3-interacting proteins that met a cut-off criterion of confidence greater than 95% and were absent in controls (Appendix A). The proteins identified by AP-MS were categorized into broad functional classes using PANTHER [41] (Figure 2a and Appendix A). Gene ontology (GO) enrichment and over-representation profiles of the dGOLPH3 interactome were analyzed using GOrilla [42] and PANTHER [41], respectively, and revealed a significant enrichment of proteins involved in cytokinesis and vesicle-mediated trafficking (Figure 2b; Appendix A).

### 3.2. Profiling dGOLPH3 Interactors Reveals an Enrichment of Membrane Trafficking Proteins

dGOLPH3-RFP pulled down well-established interactors of GOLPH3/Vps74p, which include coat protein complex I (COPI) subunits and the phosphatidylinositol 4-phosphatase Sac1 ([7,12,43,44]; Table 1, Appendix A).

Co-IP and GST pull-down experiments further validated the association of dGOLPH3 with COPI subunits (Figure 3a–d).

Consistent with the known requirement of GOLPH3 proteins for Golgi architecture maintenance and vesicular trafficking [1,2,3,4,5,17], the dGOLPH3 interactome comprises vesicle coat proteins, Rab GTPases, proteins of the tethering and fusion machinery and endocytic trafficking regulators (Table 1). Listed among the vesicle transport proteins are not only the COPI subunits but also the COPII proteins Sec31, Sec23, and Sec24CD, suggesting the role of dGOLPH3 in controlling Endoplasmic Reticulum (ER) to Golgi trafficking (Table 1, Appendix A). The binding of dGOLPH3 to Sec31 was further validated by using GST pull-down (Figure 3e). The Rab family GTPases that bound to dGOLPH3 have been implicated in multiple steps of intracellular trafficking and tethering (Table 1, Appendix A, [45]). Among these Rab GTPases, Rab1, Rab11, and Rab5 had already been identified as molecular partners of *Drosophila* and/or human GOLPH3 in previous studies [4,27,46,47]. Rab1 controls ER to Golgi and intra-Golgi trafficking whereas Rab5 and Rab11 regulate endocytic trafficking [4,27,37]. Rab8, Rab10 and Rab14 regulate post-Golgi trafficking from the *trans*-Golgi network to the plasma membrane [48]. Rab32 has been associated with vesicle trafficking through lysosome and is required for autophagy and lipid storage [49,50]. Co-IP experiments further validated binding of GOLPH3 to Rab8 and Rab10 in *Drosophila* testes (Figure 4a).

We also identified the *Drosophila* ortholog of human COG7 (namely Cog7), a subunit of the Conserved Oligomeric Golgi (COG) complex, which plays a pivotal role in tethering retrograde vesicles that traffic within the Golgi and between the endosomes and the Golgi. Further evidence of COG-dGOLPH3 association was provided in our previous studies [27,38]. Of interest is also the presence in the dGOLPH3 interactome of Sec22 and Slh, which both regulate membrane fusion events. Sec22, a v-SNARE (vesicle soluble N-ethylmaleimide-sensitive factor attachment protein receptor), is necessary in promoting efficient membrane fusion in the cis-Golgi and in the contact sites between the ER and the plasma membrane [51,52]. Slh is the *Drosophila* homolog of human Sly1, a Sec1/mammalian Unc-18 (SM) protein which functions at the ER-Golgi to regulate SNARE complex assembly and membrane fusion [53,54,55,56]. Further validation of Sec22-dGOLPH3 interaction was obtained by Co-IP experiments (Figure 4b). 

The candidate dGOLPH3 partners regulating protein transport include Sec63 and Srp54k, which are components of the signal recognition particle (SRP), the multimeric ribonucleoprotein machine that, along with its conjugate SRP receptor, controls targeting of secretory proteins to the rough ER [57]. Among additional partners, we also found *Drosophila* Gilgamesh (Gish), a plasma membrane-associated casein kinase that has been involved in the maintenance of germline stem cell sperm individualization in testes [58,59]. Gish also regulates polarized Rab11-vesicle trafficking during trichome formation [60]. In accordance with our previous data [4], we found evidence of the association between dGOLPH3 and the clathrin heavy chain subunit (Chc) (Table 1, Appendix A). Besides Chc, other dGOLPH3-partners involved in clathrin-mediated endocytic trafficking are Shibire and SH3PX1, that are, respectively, the *Drosophila* orthologs of dynamin and Sorting nexin 9 (Snx9) (Table 1, Appendix A). GST pull-down experiments further provided experimental validation of Shibire and SH3PX1 as dGOLPH3 protein partners (Figure 5a,b).

### 3.3. The dGOLPH3 Interactome Reveals Functions in Several Glycosylation Pathways

Experimental data from both yeast and human cultured cells revealed that GOLPH3 controls COPI-mediated Golgi trafficking of several Golgi glycosyltransferases required for N- and O-glycosylation [7,8,12]. Our results suggest that dGOLPH3 might bind to glycosyltransferase enzymes that control multiple glycosylation pathways such as N- and O-linked glycan synthesis and glycosylphosphatidylinositol (GPI) anchor processing (Table 1, Appendix A). Four proteins in the list of dGOLPH3 interactome, CG6790, CG5342, CG4907 and PIG-T, are predicted to be involved in GPI-anchor biosynthesis [61]. In the context of N-glycosylation, the dGOLPH3-interactome indicates an association with proteins involved in the early steps of N-glycosylation. OstΔ is a subunit of the oligosaccharyl transferase (OST) complex that catalyzes the initial transfer of Glc3Man9GlcNAc2 from dolichol-pyrophosphate to an asparagine residue within an Asn-X-Ser/Thr consensus motif in nascent polypeptides [62]. Uridine diphosphate (UDP)-glucose:glycoprotein glucosyltransferase plays a crucial role in glycoprotein quality control in the ER [63,64]. The other two proteins involved in N-glycosylation are the *Drosophila* orthologs of human alpha-1,2-mannosyltransferase (ALG11) and phosphomannomutase type 2 (PMM2) [65,66,67,68,69,70]. ALG11 controls the addition of the first alpha-1,2-linked mannose residues to growing linked-oligosaccharide [66]. Human PMM2 catalyzes the second step in the mannose pathway, which converts mannose-6-phosphate to mannose-1-phosphate, the precursor of GDP-mannose [67,68,69,70].

Our results implicate dGOLPH3 in the synthesis of mucin-type O-glycans (initiated by GalNAc-Ser/Thr) and glycosaminoglycan (GAG) chains. Consistent with the previous report from Chang and coauthors [16], we found that dGOLPH3 interacted with the exostosin Brother of tout-velu (Botv), a glucuronyl-galactosyl-proteoglycan 4-alpha-N-acetylglucosaminyltransferase required for heparan sulfate proteoglycan synthesis. Additional molecular partners include Pgant5 and Pgant7, which display N-acetylgalactosaminyltransferase activity required to initiate mucin-type O-glycosylation [71].

### 3.4. dGOLPH3 Partners Control Lipid Homeostasis and Golgi Architecture

Along with the phosphatidylinositol 4-phosphatase Sac1, other proteins that bound GOLPH3 have a role in lipid homeostasis (Table 1, Appendix A). The protein Small wing (Sl) is a phosphatidylinositol-specific phospholipase type C [72] that catalyzes the hydrolysis of phosphatidylinositol (4,5) bisphosphate into two second messengers, inositol 1,4,5-trisphosphate (IP3) and diacylglycerol (DAG). Multi-substrate lipid kinase (Mulk) is the *Drosophila* ortholog of human Acylglycerol kinase (AGK) ceramide kinase which reports to phosphorylates ceramide and acts in Wnt-mediated migration of primordial germ cells [73]. PAPLA1 enzymes cleave the ester bond at the Sn-1 position of Phosphatidic acid to produce lysophosphatidic acid, a bioactive phospholipid that mediates several signaling functions [74,75]. The genome of *Drosophila* encodes a unique PAPLA1 enzyme, while the mammalian PAPLA1 family consists of three members: DDHD1, DDHD2 (or KIAA0725p), and the SEC23 interacting protein (SEC23IP) [76,77,78]. *Drosophila* PAPLA1, by interacting with the COPII proteins Sec23 and Sec31, regulates ER to Golgi transport and glycosylation of Rhodopsin 1, an N-glycosylated G-protein coupled receptor [77]. 

The *Drosophila* Bond protein is a member of the Elovl family of enzymes that is required for the elongation of very-long-fatty-acids, commonly found in sphingolipids and essential for sphingolipid function [79,80]. Similarly to dGOLPH3, Bond has been involved in spermatocyte cytokinesis [80]. Moreover, both PAPLA1 and Bond proteins are required for sperm individualization during *Drosophila* spermatogenesis [77,81]. Among the proteins involved in lipid metabolism we found the very-long-chain enoyl-CoA reductase Sc2, required for very long-chain fatty acid biosynthetic process [82] and the Sphingosine-1-phosphate lyase Sply, which catalyzes the conversion of sphingosine-1-phosphate to ethanolamine phosphate and a fatty aldehyde [83]. The interactome of dGOLPH3 comprises several proteins required for Golgi structure and function. The ubiquitin-selective AAA-ATPase valosin-containing protein (VCP) (also known as Transitional endoplasmic reticulum 94, TER94) controls the cell-cycle-dependent Golgi fragmentation/assembly as well as the ubiquitin-proteasome system [84]. Ergic53 is the *Drosophila* ortholog of mammalian ER-Golgi intermediate compartment-53 (ERGIC-53), also known as p58 and lectin mannose binding 1 (LMAN1), a type I transmembrane protein containing a mannose binding domain that has been established as a marker of the ER-Golgi intermediate compartment (ERGIC or IC) [85,86]. Mammalian P58/ERGIC-53/LMAN1, by interacting with COPI and COPII coats, operates as a cargo receptor and a recycling protein at the ER-Golgi interface [87,88] and is essential for maintaining the architecture of ERGIC and Golgi [86]. 

### 3.5. dGOLPH3 Partners Regulate Cell Cycle Progression and Cell Signaling

Consistent with the role of GOLPH3 in cell division and proliferation, many proteins identified in our AP-MS experiments are master cell-cycle regulators and/or signaling proteins including phosphatases and kinases. Protein phosphatase 2A at 29B (Pp2A-29B) and Twins are, respectively, the structural A subunit and the regulatory subunit of PP2A [89], a heterotrimeric serine/threonine protein phosphatase that controls several cellular processes including cell cycle progression [90,91], stress-induced autophagy [92], microtubule orientation [93], centrosome duplication [94,95,96], chromosome segregation [97], actin dynamics, dendritic pruning [98] and cytokinesis [99,100].

*Drosophila* Alphabet is a metal-dependent serine/threonine phosphatase of the PP2C family, closely related to mammalian PP2Cα/β isoforms, that acts as a negative regulator of RAS/MAPK signaling [101,102].

The Mitogen-activated protein kinase kinase (MAPKK) 4 is predicted to have a Jun kinase kinase activity [103]. In accordance with our previous work [22], we provide evidence for the interaction between dGOLPH3 and the regulatory light chain Spaghetti Squash (Sqh) of non-muscle Myosin II, a structural component of the actomyosin contractile ring during cytokinesis. The *Drosophila* GTP binding proteins, Rac and Cdc42, are also key regulators of actin cytoskeleton organization and have been involved in vesicle trafficking, cell polarization and JNK kinase activation [104,105,106,107,108] Moreover, studies in mammalian cell culture systems have shown that coatomer-bound Cdc42 regulates actin assembly, dynein recruitment and bidirectional transport at the Golgi [109,110,111]. Co-IP experiments from testes expressing GFP-tagged Rac1 and Rac2 further validated binding of dGOLPH3 to Rac2 GTPase (Figure 4b).

Among the proteins identified in our AP-MS experiments, at least three proteins are involved in the TOR signaling pathway. *Drosophila* 14-3-3 ζ and translationally controlled tumor protein (Tctp) regulate TOR signaling through Rheb (Ras homolog enriched in brain), [112,113], a Ras-related GTPase that promotes TOR protein kinase activation [114,115,116,117]. In addition, *Drosophila* LST8 is the ortholog of mammalian Lst8 (MLST8), the only conserved TOR-binding protein that is a common partner for both TORC1 and TORC2 complexes and essential for *Drosophila* TORC2 activity [118,119,120].

### 3.6. dGOLPH3 Interactors Suggest an Involvement in the Assembly or Organization of Ciliary and Flagellar Axonemes

Many of the candidate partners of dGOLPH3 have been implicated in motile cilia assembly and/or function, suggesting that GOLPH3 might have a role in this process (Table 1, Appendix A). *Drosophila CG9313*, *CG3121*, *CG31803*, and *CG10859* encode the *Drosophila* orthologs of dynein axonemal intermediate chain 1, radial spoke head component 4A, radial spoke head component 9, dynein axonemal intermediate chain 2, that are essential constituents of ciliary and flagellar axonemal structures [121]. Remarkably, the human orthologs of these proteins have been involved in primary ciliary dyskinesia and Kartagener syndrome, rare autosomal recessive genetic disorders affecting motile cilia function and characterized by chronic respiratory infections and defects in male fertility [122]. Cut up (Ctp) and Cytoplasmic dynein light chain 2 (Cdlc2) belong to the dynein light chain family and exhibit dynein intermediate chain binding activity [123,124,125]. Defective transmitter release (Dtr) protein is the *Drosophila* ortholog of human dynein axonemal assembly factor 1 [121], a protein that is required for the stability of the ciliary structure and involved in cytoplasmic preassembly of dynein arms [126]. The kinesins Kinesin-like protein at 10A (Klp10A) and Kinesin-like protein at 59D (Klp59D) and Centrosomal protein of 97 kDa (CEP97) are also required for sperm ciliogenesis [127,128,129,130].

### 3.7. dGOLPH3 Interacts with Drosophila Ataxin-2 and Fragile Mental Retardation Protein

The identification in the dGOLPH3 interactome of Ataxin-2 (dAtx2) and dFmr1, two proteins that have been involved in nervous system functions and polyglutamine diseases, is particularly noteworthy ([131,132,133]; Table 1, Appendix A). dAtx2 and dFmr1 are, respectively, the *Drosophila* orthologs of human Ataxin 2 (ATX2) and fragile mental retardation protein (FMRP), two RNA binding proteins implicated in synaptic plasticity and neuronal translational control [132,134,135,136] Mutations in the human *ATXN2* gene have been linked to type-2 spinocerebellar ataxia (SCA2, [137,138,139] and a form of amyotrophic lateral sclerosis [131]. Mutations in human *FMRP* cause fragile X syndrome, characterized by mental retardation and autism, accompanied by gonadal defects [140,141,142,143,144,145,146]. Binding of dGOLPH3 to dAtx2 and dFmr1 was further validated by using GST pull-down and Co-IP experiments (Figure 6a,b).

Immunostaining of primary spermatocytes for dAtx2-HA and dFmr1 showed that these proteins co-localize in the cytoplasm of premeiotic spermatocytes as expected from their described roles in translational control (Figure 7a).

Immunostaining of testes with anti-dFmr1 and anti-dGOLPH3 antibodies showed that dFmr1 is enriched at the midzone and overlaps with dGOLPH3-enriched organelles at the astral membranes of dividing spermatocytes (Figure 7b). Recent data have shown that dAtx2 interacts and functions with dFmr1 in neuronal translational control to mediate long-term olfactory habituation [133]. We further validated the interaction between dGOLPH3/dFmr1 and dFmr1/dAtx2 in fixed spermatocytes using a proximity ligation assay (PLA, Figure 8).

Taken together, our results indicate that dFmr1 interacts with both dGOLPH3 and dAtx2 in the cytoplasm of male meiotic cells.

## 4. Discussion

### 4.1. The dGOLPH3 Interactome Reveals an Enrichment in Vesicle-Mediated Trafficking and Cytokinesis Proteins

In this paper, we report the first comprehensive analysis of the interactome of GOLPH3 protein. We have exploited the advantages of *Drosophila* spermatogenesis, which offers a well-suited model system for dissecting membrane trafficking pathways and their role in cytokinesis and cell differentiation [23,24,25,26,27]. Consistent with our previous findings that dGOLPH3 controls membrane trafficking during cytokinesis [4,21,22,27], the dGOLPH3 interactome revealed an enrichment of proteins involved in cytokinesis and vesicle-mediated trafficking. Importantly, we have identified well-established molecular partners of GOLPH3/Vps74 such as COPI subunits, Sac1 and Rab1 [7,12,43,44,46,147]. Among the novel molecular interactors of dGOLPH3, we found vesicle coats, small GTPases of the Rab family, and tethering and fusion factors, indicating roles in both secretory and endocytic trafficking pathways. We showed that dGOLPH3 bound Rab8 and Rab10 proteins that regulate post-Golgi trafficking from the *trans*-Golgi network to the plasma membrane [48]. Rab8 functions with Rab10 and Rab14 in GLUT4 cycling [148]. Moreover, both Rab8 and Rab10 contribute to ciliogenesis [148,149]. dGOLPH3 also bound other Rab GTPases involved in controlling endocytic trafficking. Importantly, distinct Rab proteins localize at specific membrane-bound compartments and act in concert with different phosphoinositides to regulate all the vesicular trafficking pathways [150]. Although GOLPH3 localization to the Golgi membranes depends on PI(4)P, both human and *Drosophila* GOLPH3 proteins were able to bind PI(3)P and PI(4,5)P2 in lipid-binding assays [4,13]. Moreover, by using surface plasmon resonance, Wood and co-authors [13] demonstrated that the human GOLPH3 binds PI(3)P with a mere threefold affinity compared with PI(4)P. Thus, GOLPH3 proteins might associate with either PI(4)P-vesicles or PI(3)P-enriched endosomes and regulate secretory and endosomal membrane dynamics in concert with specific Rab GTPases, during interphase and cytokinesis. 

In the context of the GOLPH3-controlled vesicle trafficking in cytokinesis, a vast amount of literature has discussed the role of endocytosis and endocytic recycling pathways during furrow ingression and the final steps of cytokinesis in model organisms and mammalian cultured cells [151,152,153]. At least two endocytic Rab GTPases, namely Rab11 and Rab35, control distinct endocytic recycling pathways required for completion of cytokinesis in mammalian cells [154,155,156,157,158,159]. Consistent with our previous studies on fly spermatocytes [20,27], our AP-MS experiments identified Rab11 (but not Rab35) as a molecular partner of dGOLPH3. In *Drosophila melanogaster* Rab11 is essential for cytokinesis of S2 cells and spermatocytes [159,160,161] as well as for furrow ingression during embryonic cellularization [162,163]. In dividing spermatocytes, Rab11 concentrates to the cleavage furrow, together with its effector Nuclear fallout [160,161], providing an essential function for contractile ring constriction and furrow ingression [160]. Data from the Brill group showed that the *Drosophila* type III PI 4-kinase four wheel drive (Fwd), localizes to the Golgi of male meiotic cells, recruits Rab11 to the Golgi complex and is required for the accumulation of PI(4)P-vesicles co-localizing with Rab11 at the cell equator of dividing spermatocytes [161]. However, because Fwd does not localize to the cleavage furrow, targeting of PI(4)P and Rab11 vesicles to the equatorial site depends on dGOLPH3 function [4]. Thus, data in this paper together with our previous analyses [4,21,22] suggest a model whereby the PI(4)P effector dGOLPH3 forms a complex with Rab11 and myosin II and coordinates contractile ring assembly with phosphoinositide signaling and vesicle trafficking during cytokinesis. 

In contrast to fly spermatocytes, in mammalian cells, the function of Rab11 is dispensable for cleavage furrow ingression but required for abscission [154,155,156,158]. Further studies on mammalian cells should clarify whether GOLPH3 functions with the Rab11 pathway to promote actin clearance and ESCRT-III dependent abscission in late cytokinesis [158].

The endocytic trafficking proteins, clathrin and dynamin, have been also involved in cytokinesis in several model organisms [164,165,166]. Consistent with our previous findings [4,21], the interactome of dGOLPH3 indicates a potential molecular interaction between GOLPH3 and Chc, that is also consistent with the presence of a putative clathrin-binding motif, LLDLD, in the GOLPH3 amino acid sequence [4]. During clathrin-mediated endocytosis (CME), the GTPase dynamin assembles into a collar-like structure at the necks of clathrin coated picks and directs membrane fission and vesicle release [167,168,169]. Functions of dynamin require its association with numerous Src homology 3 domain (SH3)-containing proteins including Sorting nexin 9 (SNX9, [170,171].

Our findings demonstrate that dGOLPH3 interacts with Shibire and SH3PX1 that are, respectively, the *Drosophila* orthologs of human dynamin and SNX9 proteins [172,173,174]. Mammalian SNX9 proteins consist of three paralogs, SNX9, SNX18, and SNX33, which have been involved in endosomal vesicle sorting [170,175,176,177], endosomal retrograde trafficking [178] and actin cytoskeleton remodeling [179,180,181]. Besides the SH3 domain, SNX9 and SH3PX1 share a phox (phagocyte oxidase) homology domain (PX), a low complexity (LC) region and a Bin-Amphiphysin-Rvs (BAR) domain [182,183]. The N-terminal SH3 domain of SNX9 is required for binding the proline rich domain (PRD)-containing proteins, such as dynamins, WASP and N-WASP [170,184,185]. The LC region mediates binding with ARP2/3 complex [186]. The C-terminal PX and BAR domains are involved in binding to phosphoinositides and control curvature sensing and dimerization [176,180,187,188]. Similar to dGOLPH3, SNX9 protein connects actin polymerization with membrane remodeling and vesicle formation [189]. 

Importantly, human SNX9 subfamily proteins are required for accumulation of active myosin II at the cleavage site and normal furrow ingression during cytokinesis of HeLa cells [179]. Further work will clarify whether SH3PX1/SNX9 protein cooperates with GOLPH3 to regulate membrane remodeling and actomyosin dynamics during cytokinesis. 

Although most studies on membrane trafficking during cytokinesis have revealed the essential role of the Golgi and endocytic pathways proteins, it has been suggested that the ER might provide an important membrane storage within the dividing cells [190]. Consistent with a possible role of the ER in cytokinesis, a proteomic analysis of purified midbodies isolated from Chinese hamster ovary cells led to the identification of both ER-resident proteins and proteins involved in ER to Golgi traffic including Sec13, endoplasmin, Sec23, Sec31 and COPI [191]. Moreover, time-lapse fluorescence analysis of the ER disulfide isomerase GFP chimera protein revealed pronounced reorganization during cytokinesis and a redistribution of this protein to the spindle poles and the spindle equator of *Drosophila* dividing cells [192,193]. The interactome of dGOLPH3 comprises the COPII proteins Sec31, Sec23 as well as ERGIC-53 and Sec22 proteins, which are known to operate at the level of the ERGIC/cis-Golgi. Additionally, consistent with our previous findings [27], data in this paper confirm the interaction between dGOLPH3 and Rab1, which controls ER to Golgi and intra-Golgi trafficking. Future studies will be required to explore the functional dependence between dGOLPH3 and ER/ERGIC/cis-Golgi proteins and the possible implications of these interactions during cytokinesis.

### 4.2. dGOLPH3 Interacts with Proteins Required for Protein Glycosylation and Lipid Homeostasis

It has been amply demonstrated that GOLPH3 proteins control COPI-mediated Golgi trafficking of specific Golgi glycosyltransferases [7,8,10,12]. Our findings suggest that dGOLPH3 is required for N- and O-linked glycan synthesis and glycosylphosphatidylinositol (GPI) anchor processing. Importantly our results indicate a role of dGOLPH3 in the biosynthesis of glycosaminoglycan chains, that was also reported by Chang and coauthors [16]. Moreover, our results implicate dGOLPH3 in the synthesis of mucin-type O-glycans. It has been proposed that the oncogenic properties of human GOLPH3 might be correlated with defects in protein glycosylation [5]. Indeed, altered glycoprotein glycosylation represents a hallmark of cancer and in particular aberrant mucin-type O-glycans have an important role in cancer pathogenesis as they affect the adhesive properties of the neoplastic cells and promote cell invasion and tumor metastasis [194]. 

The interactome of dGOLPH3 indicates an important role in lipid metabolism and signaling that correlates with tumorigenesis and a variety of human genetic diseases [195]. In this context, several molecular interactors of dGOLPH3 are required for the synthesis of sphingolipids and other signaling lipids including lysophosphatidic acid and DAG. These data support the model whereby PI(4)P-GOLPH3 exerts a key function to coordinate lipid homeostasis with vesicle trafficking and glycosylation at the Golgi. 

### 4.3. The dGOLPH3 Interactome Indicates Molecular Targets that Might Be Relevant for Therapeutic Intervention in Cancer and Other Neurological Diseases

It has been suggested that human GOLPH3 and Golgi alterations might have a potential role in the pathophysiology of neurological disease [196]. Remarkably, Golgi fragmentation of specific groups of neurons is an early preclinical event in many neurodegenerative diseases, preceding pathological symptoms [197]. Our findings that dGOLPH3 interacts with dFmr1 and dAtx2 suggest a link with polyglutamine diseases. Studies on the members of Ataxin-2 from human cells and model organisms indicated a conserved role of ATX2 proteins in regulating of mRNA stability and translation [133,134,198,199,200,201,202]. ATX2 has been also involved in endocytic trafficking [203,204] and has been localized to ER membranes and Golgi [205,206]. Recent studies provided evidence for a conserved role of Atx2 in ER dynamics and structure in *C. elegans* as well as in *Drosophila* embryos and cultured neurons suggesting a possible mechanism that involves vesicle trafficking in SCA2 disease [207]. Mammalian FMRP and dFmr1 are mainly localized in the cytoplasm, where they bind specific mRNAs acting as translation regulators [133,136,208,209,210]. FMRP proteins also interact with components of the microRNA and Piwi-interacting RNA pathways [211,212,213,214,215]. Recent data have shown that dAtx2 interacts and functions with dFmr1 in neuronal translational control to mediate long-term olfactory habituation [133]. Our results indicate that dFmr1 forms a complex with both dAtx2 and dGOLPH3 in male meiotic cells. In addition, dFmr1 protein is enriched at the poles and in the midzone of the dividing spermatocytes, suggesting a potential role in cytokinesis. We speculate that dFmr1 protein might control localization and transport of specific mRNAs to midzone microtubules where they are locally activated during cytokinesis. A similar regulatory process occurs in neurons where mRNAs, together with the machinery for RNA translation, are transported from the cell body to synapses where they are locally translated [216]. Moreover, cytokinesis of early *C. elegans* embryos requires ATX2 function, which controls a molecular mechanism required to target and maintain the kinesin ZEN-4 to the spindle midzone through the posttranscriptional regulation of PAR-5 [217]. Consistent with a role of dFmr1 in cytokinesis, several mRNA/protein targets of dFmr1, so far identified, are involved in actin cytoskeleton remodeling. dFmr1 binds Cytoplasmic FMRP Interacting Protein (CYFIP), the fly ortholog of vertebrate FMRP interactor CYFIP1, that is part of the WAVE regulatory complex that regulates actin polymerization [218,219]. In addition, the function of dFmr1 in dendritic development depends on the small GTPase Rac1 and the Rac1-encoding mRNA is present in the Fmr1-messenger ribonucleoprotein complexes [209]. Finally, dFmr1 controls actin cytoskeleton dynamics in *Drosophila* neurons by binding the mRNA of the *Drosophila* profilin homolog *chickadee* and regulating Profilin protein expression [210]. Although the relationship between the dGOLPH3 and the dFmr1-mediated mRNA transport remains to be determined, proteins of the COPI vesicle complex interact with specific mRNAs and disruption of the COPI complex results in mis-localization of RNAs in human neurons [220,221]. Moreover, COPI α binds a specific set of mRNAs that overlaps with FMRP-associated mRNAs which encode proteins that are known to localize to the plasma membrane and cytoskeleton [222]. Uncovering the molecular mechanisms that involve dGOLPH3, and dFmr1 in the dynamics of RNAs during cell division will further our understanding of diseases of the nervous system. Moreover, recent studies indicate the involvement of human FMRP in different cancer types including breast cancer and melanoma [223,224]. Thus, investigating the functional dependence between GOLPH3 and FMRP will be also important in the light of a therapeutic strategy in human cancer. 

The Co-IP of dGOLPH3 is also enriched with proteins controlling cell cycle progression and cell proliferation including protein kinases and phosphatases. A significant finding in our work is the interaction of dGOLPH3 with proteins that are known to play a role in the TOR kinase signaling pathway. Importantly, in the context of cancer pathogenesis, human GOLPH3 function has been associated with enhanced AKT/mTOR signaling, although the precise biochemical basis for its activity remains to be determined [17]. Importantly, findings in this paper suggest that GOLPH3 might form a complex with proteins that have been involved in the TOR signaling pathway: Tctp, 14-3-3 ζ [112,113] and the conserved TOR-binding protein LST8 [118,119,120]. Further work will clarify whether the association between GOLPH3 and these proteins can impact on the TORC1 and TORC2 complexes and consequently on cell growth and proliferation. 

## Figures and Tables

**Figure 1 cells-10-02336-f001:**
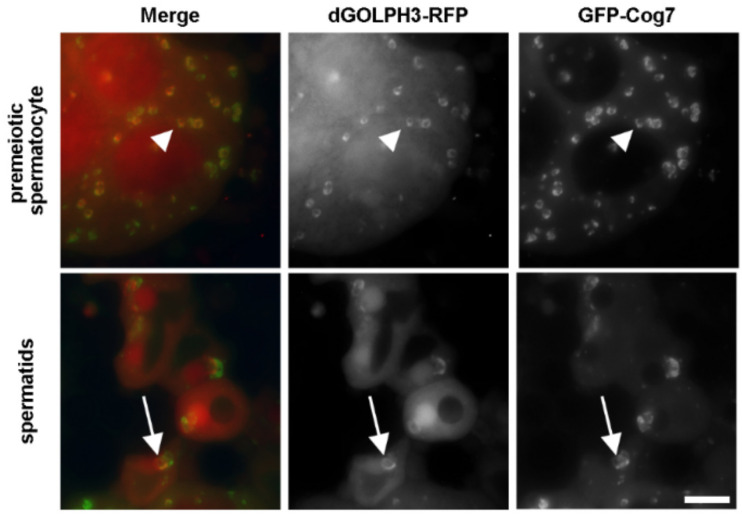
dGOLPH3-RFP localizes to the multiple Golgi organelles in primary spermatocytes and to the acroblast in spermatids. Fluorescence micrographs of live squashed spermatocytes and spermatids expressing dGOLPH3-RFP and GFP-Cog7. Arrowheads point to Golgi stacks; arrows point to the acroblast. Bar, 10 μm.

**Figure 2 cells-10-02336-f002:**
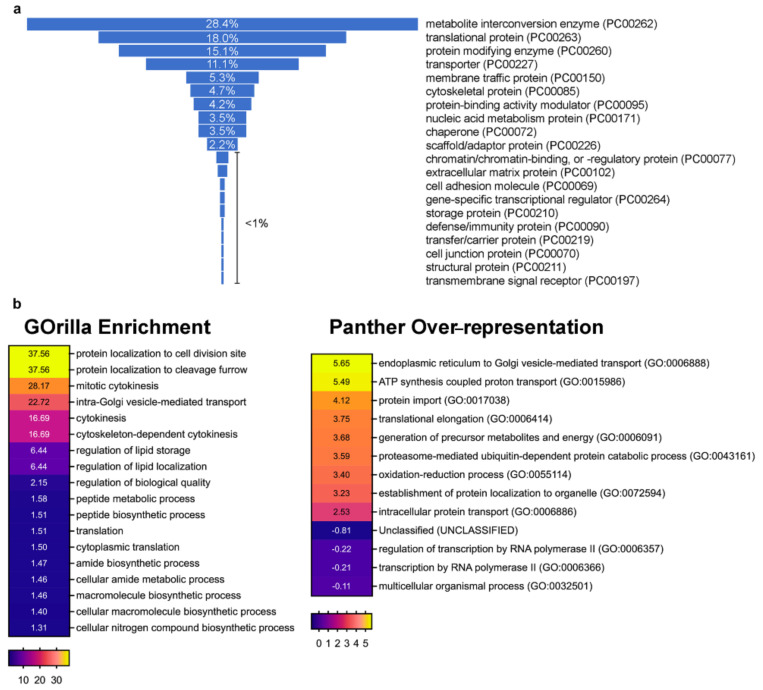
Protein Classes and GO analyses. (**a**) Funnel graph showing the classes of proteins identified in our AP-MS experiments and classified according to PANTHER database [41]. The percentage for each class is indicated. See also Appendix A for details. (**b**) Heat maps showing the GO annotation enrichment profiles of the dGOLPH3 interactome. GO enrichment profiles were analyzed using GOrilla tool [42] under the category “process” and PANTHER database [41] under the category “GO-slim biological process”. Over-represented/enriched GO terms are shown in different color shades according to their fold enrichment as indicated in the color scale bar at the bottom; actual fold enrichment values are shown within the heat map (see Appendix A for *p*-values). For simplicity and to improve visual representation, for the PANTHER over-representation analysis only the headings for each GO-slim biological process are shown in the graph, while the full results are reported in the Appendix A. Only results for Bonferroni-corrected analysis (*p* < 0.05) were considered.

**Figure 3 cells-10-02336-f003:**
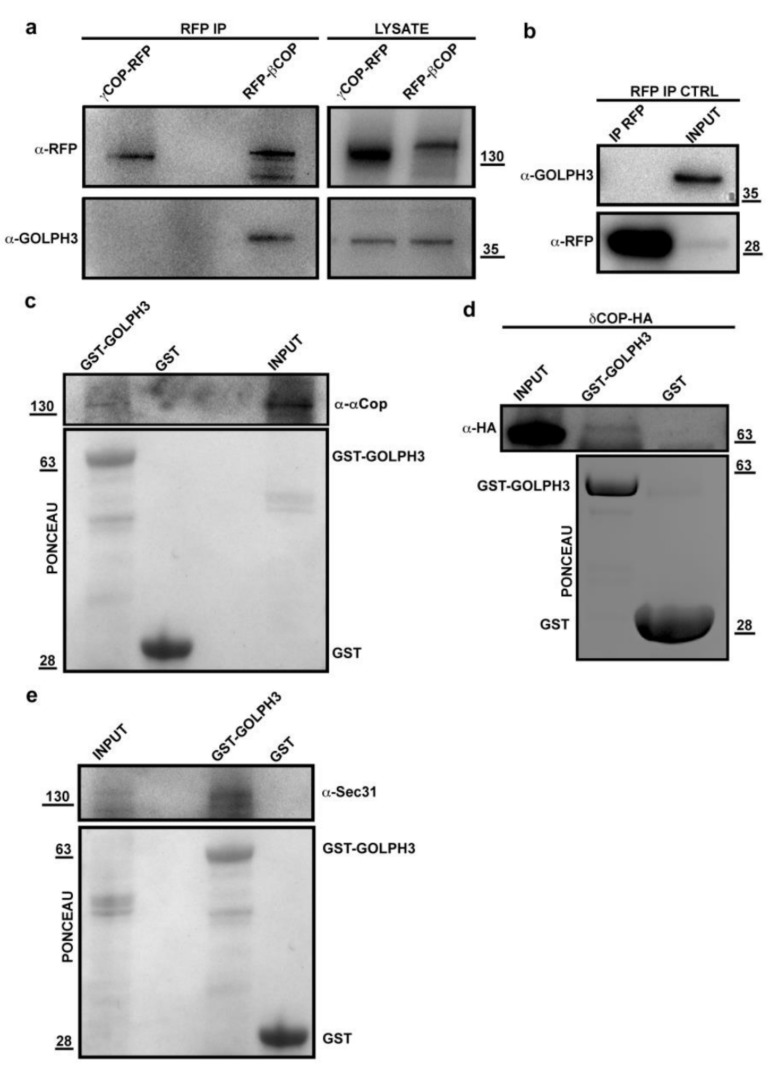
dGOLPH3 interacts with COPI subunits and Sec31 protein. (**a**,**b**) dGOLPH3 protein coprecipitated with RFP–βCOP but not with γCOP-RFP and RFP. Protein extracts from testes expressing RFP–βCOP, γCOP-RFP (**a**) and RFP (RFP IP CTRL) (**b**), were immunoprecipitated with RFP-trap beads (α-RFP) and blotted for either RFP or dGOLPH3. 4% of the total lysates and one third of the IP were loaded and probed with the indicated antibodies. (**c**–**e**) GST pull-down to test dGOLPH3 interaction with αCOP (**c**), δCOP (**d**) and Sec31 (**e**) proteins. (**c**) Bacterially expressed GST and GST-dGOLPH3 were purified by Gluthatione-Sepharose beads, incubated with testis protein extracts from Oregon-R males and blotted for αCOP protein. (**d**) Bacterially expressed GST and GST-GOLPH3 were purified by Gluthatione-Sepharose beads, incubated with testis protein extracts from males expressing δCOP-HA. (**e**) Bacterially expressed GST-dGOLPH3 and GST purified by Gluthatione-Sepharose beads were incubated with testis protein extracts from Oregon-R males and blotted for Sec31. Ponceau staining in (**c**–**e**) is shown as a loading control. 2% of the input and 25% of the pull-downs were loaded and probed with the indicated antibody. Molecular masses in (**a**–**e**), expressed in kilodaltons.

**Figure 4 cells-10-02336-f004:**
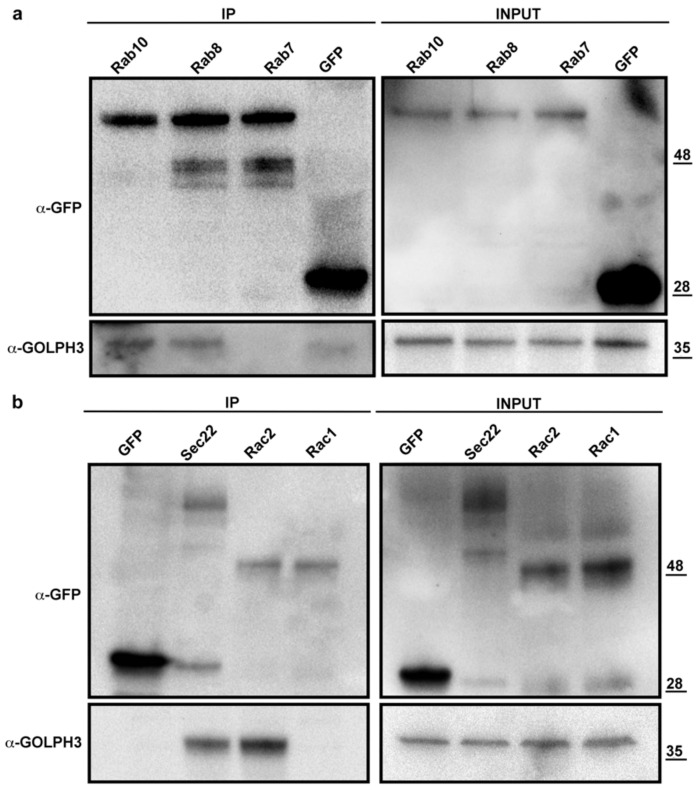
dGOLPH3 interacts with vesicle trafficking and cytoskeleton proteins. (**a**) Protein extracts from testes expressing either GFP or the indicated YFP-tagged Rab protein were immunoprecipitated with GFP traps and blotted to detect either GFP/YFP or GOLPH3. 4% of the total lysates and one third of IP were loaded and probed with the indicated antibodies. (**b**) Protein extracts from *Drosophila* testes expressing GFP (control) and GFP tagged Rac1, Rac2 and Sec22 were immunoprecipitated with GFP trap and blotted to detects either GFP or GOLPH3. 4% of the total lysates and one third of IP were loaded and probed with the indicated antibodies. Molecular masses in (**a**,**b**) are expressed in kilodaltons.

**Figure 5 cells-10-02336-f005:**
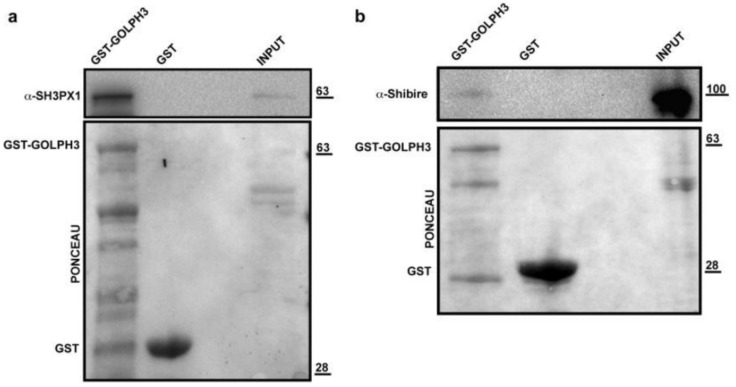
dGOLPH3 interacts with endocytic trafficking proteins. GST (control) and recombinant GST-GOLPH3 proteins, immobilized on Gluthatione-Sepharose beads were incubated with testis protein extracts from Oregon-R males. Ponceau staining is shown as a loading control. 2% of the input and 25% of the pull-down were loaded and probed for SH3PX1 (**a**) and Shibire (**b**). Molecular masses are expressed in kilodaltons.

**Figure 6 cells-10-02336-f006:**
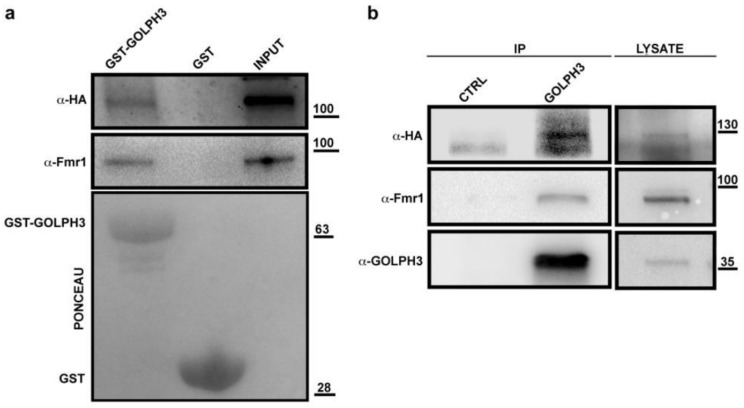
dGOLPH3 interacts with dAtx2 and dFmr1. (**a**) Bacterially expressed GST-dGOLPH3 and GST (control) were purified by Gluthatione-Sepharose beads and incubated with testis extracts expressing dAtx2-HA. Ponceau staining is shown as a loading control. A percentage of 2% of the input and 25% of pull-downs were loaded and probed with the indicated antibody. Molecular mass is expressed in kilodaltons. (**b**) Protein extracts from testes expressing dAtx2-HA were immunoprecipitated with antibodies against *Drosophila* GOLPH3 (rabbit G49139/77) and blotted with mouse anti-dGOLPH3 S11047/1/56, mouse anti-dFmr1, or mouse anti-HA. Pre-immune serum (G49139/1, from the same animal before the immunization) was used in control experiments (CTRL). Molecular mass is expressed in kilodaltons.

**Figure 7 cells-10-02336-f007:**
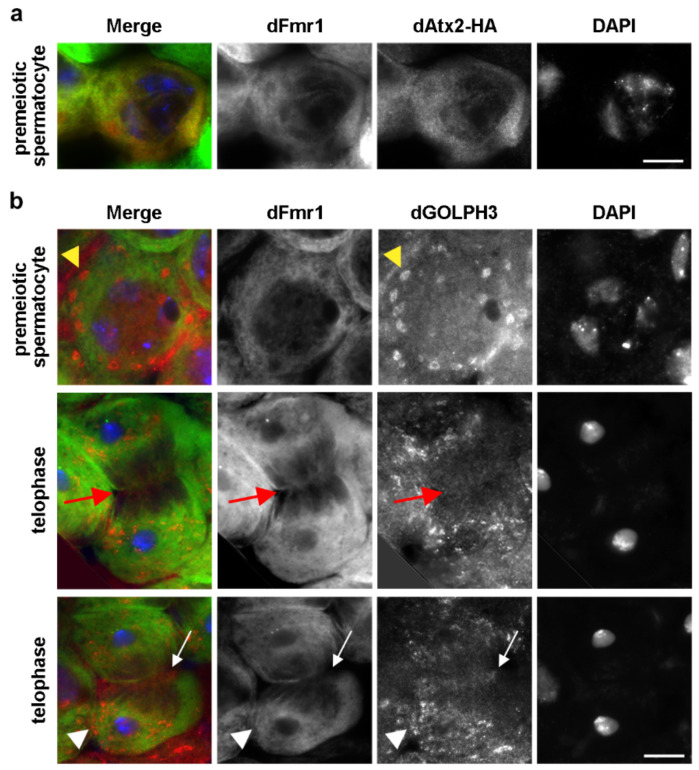
dFmr1 localization in premeiotic and dividing spermatocytes. (**a**) dFmr1 co-localizes with dAtx2 in the cytoplasm of primary spermatocytes. Testes expressing dAtx2-HA were stained for dFmr1(green), HA (dAtx2, red) and DNA (DAPI, blue). *n* = 40 spermatocytes randomly selected from images taken in five independent experiments. (**b**) Premeiotic and dividing spermatocytes were stained for dFmr1 (green) dGOLPH3 (red) and DNA (DAPI, blue). dFmr1 localizes to the cytoplasm of premeiotic spermatocytes and concentrates at the midzone (arrows) and at the astral membranes (arrowheads) at each pole of dividing spermatocytes (telophase). dGOLPH3 protein is visible in the cytoplasm and enriched in the Golgi stacks of premeiotic spermatocytes (yellow arrowheads). During telophase dGOLPH3 protein localizes to the midzone (arrows) and is enriched in puncta at the astral membranes of dividing spermatocytes (arrowheads). *n* = 40 premeiotic spermatocytes and *n* = 25 telophase spermatocytes, randomly selected from images taken in five independent experiments. Bars, 10 μm.

**Figure 8 cells-10-02336-f008:**
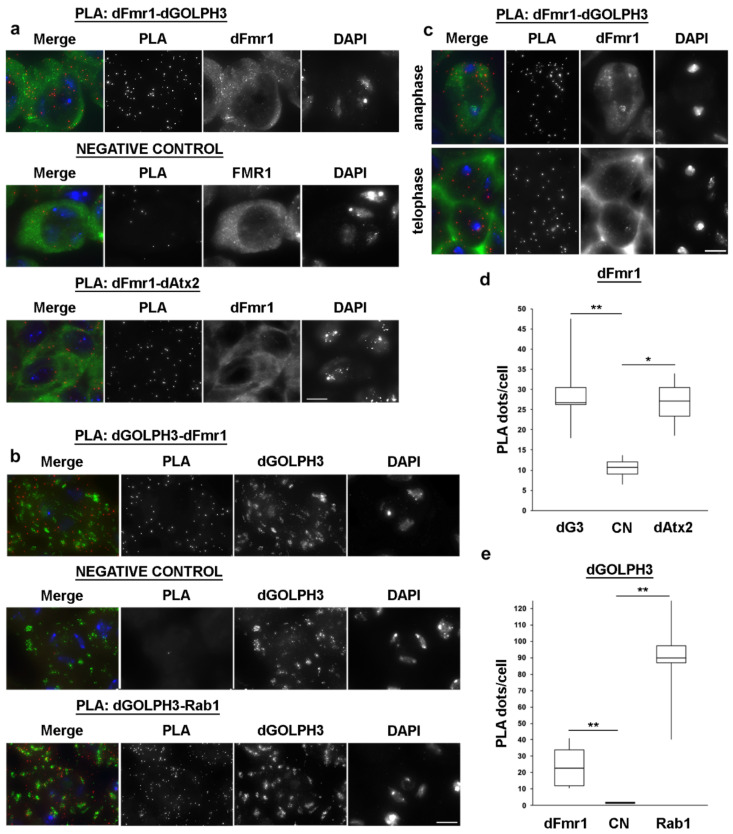
dFmr1 interacts with dAtx2 and dGOLPH3 in primary spermatocytes. (**a**,**b**) Proximity ligation assay (PLA) to visualize dFmr1-dGOLPH3 and dFmr1-dAtx2 interactions in premeiotic spermatocytes. In negative control primary anti-GOLPH3 (**a**) or anti-dFmr1 (**b**) antibodies were omitted. dGOLPH3-Rab1 interaction was used in positive control experiments. (**c**) PLA to visualize dFmr1-dGOLPH3 interaction in dividing spermatocytes. (**d**,**e**) Quantification of the PLA signals per cell was obtained as described in Materials and Methods. The box plots show the number of PLA dots per cell in dFmr1-dGOLPH3, dFmr1-dAtx2 and dGOLPH3-Rab1compared to negative controls (CN). dG3, dGOLPH3. A total of 50 cells were randomly selected from images taken in 5 independent experiments. Statistic significant differences are * *p* < 0.05; ** *p* < 0.01 (Mann-Whitney test). Bars, 10 μm.

**Table 1 cells-10-02336-t001:** Selected proteins interacting with dGOLPH3, categorized by function. The full list of hits in the dGOLPH3 interactome is shown in supplementary Appendix A. Protein identifications were verified by manual inspection of MS/MS spectra.

UniProtKD Entry ^1^	*Drosophila* Annotation Symbol ^2^	*Drosophila* Symbol ^3^	PEP ^4^
Golgi vesicle transport—GO: 0048193			
Q9W0B8	*CG7961*	αCOP	3.71 × 10^−115^
P45437	*CG6223*	βCOP	2.64 × 10^−71^
O62621	*CG6699*	β’COP	6.21 × 10^−70^
Q8I0G5	*CG1528*	γCOP	6.38 × 10^−40^
Q9W555	*CG14813*	δCOP	1.35 × 10^−28^
A1Z7J7	*CG8266*	Sec31	2.17× 10^−15^
M9PGI6	*CG7359*	Sec22	1.47 × 10^−12^
Q9VLS7	*CG8552*	PAPLA1	1.59 × 10^−10^
Q9VQ94	*CG10882*	Sec24CD	3.34 × 10^−10^
Q9VNF8	*CG1250*	Sec23	1.29 × 10^−09^
Q24179	*CG3539*	Slh	1.88 × 10^−05^
Q9Y0Y5	*CG9543*	εCOP	1.83 × 10^−04^
Q9VAD6	*CG31040*	Cog7	4.91 × 10^−03^
Rab protein signal transduction—GO: 0032482			
O18332	*CG3320*	Rab1	3.74 × 10^−69^
A1Z7S3	*CG8024*	Rab32	5.99 × 10^−23^
Q9V3I2	*CG3664*	Rab5	2.60 × 10^−22^
O18338	*CG8287*	Rab8	5.90 × 10^−06^
O15971	*CG17060*	Rab10	5.90 × 10^−06^
Q86BK8	CG4212	Rab14	1.63 × 10^−05^
O18335	CG5771	Rab11	8.83 × 10^−05^
Transport—GO: 006810			
P29742	*CG9012*	Chc	1.36 × 10^−43^
Q9VS57	*CG8583*	Sec63	1.58 × 10^−43^
Q9V3D9	*CG4659*	Srp54k	1.90 × 10^−22^
P27619	*CG18102*	Shi	1.23 × 10^−10^
Q9VEX2	*CG6963*	Gish	1.21 × 10^−05^
Q9NCC3	*CG6757*	SH3PX1	3.91 × 10^−05^
Protein glycosylation—GO: 0006486			
Q7K110	*CG6370*	OstΔ	5.06 × 10^−11^
Q09332	*CG6850*	Uggt	9.90 × 10^−10^
Q9VP06	*CG11306*	Alg11	6.42 × 10^−08^
Q9XZ08	*CG15110*	Botv	9.17 × 10^−06^
Q6WV17	*CG31651*	Pgant5	1.87 × 10^−05^
Q9VTZ6	*CG10688*	Pmm2	3.61 × 10^−04^
Q8MV48	*CG6394*	Pgant7	1.17 × 10^−02^
Lipid metabolic process—GO: 0006629			
Q9W0I6	*CG9128*	Sac1	1.43 × 10^−49^
Q9VXH3	*CG4200*	Sl	1.04 × 10^−10^
Q9VL10	*CG31873*	Mulk	4.69 × 10^−10^
Q9VGM0	*CG6790*	CG6790	4.19 × 10^−09^
Q9VZL3	*CG10849*	Sc2	1.13 × 10^−08^
Q9VGL9	*CG5342*	CG5342	2.21 × 10^−08^
Q9VCV7	*CG4907*	CG4907	3.48 × 10^−07^
Q9V7Y2	*CG8946*	Sply	2.20 × 10^−06^
Q9VCY7	*CG6921*	Bond	5.56 × 10^−04^
Q9W3G0	*CG11190*	PIG-T	3.44 × 10^−03^
Golgi organization—GO: 0007030			
Q7KN62	*CG2331*	TER94	5.55 × 10^−45^
Q7KNA0	*CG8230*	CG8230	4.24 × 10^−16^
Q9V3A8	*CG6822*	ergic53	1.23 × 10^−05^
Cell cycle—GO: 0007049			
P36179	*CG17291*	Pp2A-29B	2.55 × 10^−11^
Q9VAK1	*CG1906*	Alph	2.56 × 10^−05^
P36872	*CG6235*	Tws	2.18 × 10^−03^
P40423	*CG3595*	Sqh	4.19 × 10^−03^
Signaling—GO: 0023052			
P29310	*CG17870*	14-3-3ζ	1.36 × 10^−57^
P40792	*CG2248*	Rac1	1.01 × 10^−41^
P48554	*CG8556*	Rac2	1.01 × 10^−41^
Q9VGS2	*CG4800*	Tctp	1.71 × 10^−06^
O61444	*CG9738*	Mkk4	1.43 × 10^-04^
Q9W328	*CG3004*	Lst8	8.71 × 10^−04^
P40793	*CG12530*	Cdc42	1.72 × 10^−03^
Cilium assembly—GO: 0060271			
Q8INT5	*CG31623*	Dtr	2.96 × 10^−15^
Q9V3M9	*CG4767*	Tektin-A	4.85 × 10^−13^
Q9W1D3	*CG3121*	Rsph4a	1.12 × 10^−08^
O96860	*CG5450*	Cdlc2	1.58 × 10^−05^
Q24117	*CG6998*	Ctp	1.58 × 10^−05^
Q960Z0	*CG1453*	Klp10A	1.61 × 10^-05^
Q8T3V7	*CG31803*	Rsph9	1.72 × 10^−05^
Q9W1U1	*CG12192*	Klp59D	1.92 × 10^−05^
Q9VQV7	*CG3980*	Cep97	3.60 × 10^−05^
Q9VJY4	*CG10859*	CG10859	2.45 × 10^−04^
Q8MSJ9	*CG9313*	CG9313	1.12 × 10^−03^
Nervous system process—GO: 0050877			
Q9NFU0	*CG6203*	Fmr1	9.29 × 10^−41^
Q8SWR8	*CG5166*	Atx2	8.52 × 10^−11^

^1^ UniProtKD entry; unique and stable entry identifier. ^2^
*Drosophila* annotation symbol; Current FlyBase annotation identifier of the gene. CG, prefix for protein-coding genes. ^3^
*Drosophila* symbol; Approved *Drosophila* gene/protein symbol. ^4^ PEP; Posterior Error Probability of the identification. This value essentially operates as a *p*-value and represents the probability that the observed peptide spectrum match (PSM) is incorrect (a smaller value is more significant).

## Data Availability

Not applicable.

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
