# Peer review of "Identification of GOLPH3 Partners in Drosophila Unveils Potential Novel Roles in Tumorigenesis and Neural Disorders"

_cells, 2021, doi:10.3390/cells10092336_

Round 1
Reviewer 1 Report
This is a carefully conducted and mainly solid study on proteins interacting with the Golgi-associated oncoprotein GOLPH3. It would be acceptable for publication in Cells after the authors have considered and responded to the following points:
- In Table 1 ERGIC-53 is also listed as one of the interacting proteins under the category "Golgi organization". However, it should be noted that this protein with various names (p58/ERGIC-53/LMAN1) is a cargo receptor, which by interacting with COPII and COPI coats continuously cycles at the ER-Golgi interface. In mammalian cells the protein is enriched in the intermediate compartment (ERGIC or IC), but its localization in Drosophila cells has not been reported - at least to my knowledge. The reference for ERGIC-53 (#85) deals with the function of TANGO1 - not ERGIC-53!
- It is not quite clear to me what the arrow in the panel at the bottom right corner points to.
- Unavoidably, the description of the results (as well as the Discussion) in this type of paper becomes rather list-like. Maybe the authors should make more efforts to focus on the most essential themes.
- Cdc42 is not only linked to cell polarity and the actin cytoskeleton, but also to Golgi trafficking and COPI coats (See papers by Mark Stamnes and Victor Hsu).
- The data in Figure 7b are not quite clear. I'm not totally convinced of the preferential accumulation of GOLPH3 in the midzone (is the red signal specific?) Maybe staining for the microtubules of the intercellular bridge could give a better reference point? The PLA data in Figure 8 on the other hand appears more convincing, but does not add to the point regarding (co)localization of the proteins at the midzone.
- What is meant by "polar localization"? It seems to me that the GOLPH3 signal for instance is enriched at the sides of the nucleus facing the intercellular bridge.
- Finally, one thing that I find very puzzling is that although GOLPH3 is commonly considered as a trans-Golgi/TGN protein (e.g. Sechi et al., 2020), it interacts with Rab1 (earlier data by the authors), ERGIC-53 and Sec22, all of which (at least in mammalian cells) are known to operate at the level of the ERGIC/cis-Golgi. Maybe the authors should point out this spatial dilemma, although future studies will certainly be required to find out the answer to this question.
Author Response
Point 1:
In Table 1 ERGIC-53 is also listed as one of the interacting proteins under the category "Golgi organization". However, it should be noted that this protein with various names (p58/ERGIC-53/LMAN1) is a cargo receptor, which by interacting with COPII and COPI coats continuously cycles at the ER-Golgi interface. In mammalian cells the protein is enriched in the intermediate compartment (ERGIC or IC), but its localization in Drosophila cells has not been reported - at least to my knowledge. The reference for ERGIC-53 (#85) deals with the function of TANGO1 - not ERGIC-53!
Response 1
As requested, we have amended the manuscript to clarify that ERGIC-53 protein, listed as one of the interacting proteins under the category "Golgi organization", is a cargo receptor, which by interacting with COPII and COPI coats continuously cycles at the ER-Golgi interface (see lines 421-428, page 13). Additionally, we have provided new references (85-88) and deleted the former reference #85.
Point 2
It is not quite clear to me what the arrow in the panel at the bottom right corner points to.
Response 2
We moved the arrow in the indicated panel. We think that the ribbon-like acroblast indicated by the arrow is clear now.
Point 3
Point 1:
In Table 1 ERGIC-53 is also listed as one of the interacting proteins under the category "Golgi organization". However, it should be noted that this protein with various names (p58/ERGIC-53/LMAN1) is a cargo receptor, which by interacting with COPII and COPI coats continuously cycles at the ER-Golgi interface. In mammalian cells the protein is enriched in the intermediate compartment (ERGIC or IC), but its localization in Drosophila cells has not been reported - at least to my knowledge. The reference for ERGIC-53 (#85) deals with the function of TANGO1 - not ERGIC-53!
Response 1
As requested, we have amended the manuscript to clarify that ERGIC-53 protein, listed as one of the interacting proteins under the category "Golgi organization", is a cargo receptor, which by interacting with COPII and COPI coats continuously cycles at the ER-Golgi interface (see lines 421-428, page 13). Additionally, we have provided new references (85-88) and deleted the former reference #85.
Point 2
It is not quite clear to me what the arrow in the panel at the bottom right corner points to.
Response 2
We moved the arrow in the indicated panel. We think that the ribbon-like acroblast indicated by the arrow is clear now.
Point 3
Unavoidably, the description of the results (as well as the Discussion) in this type of paper becomes rather list-like. Maybe the authors should make more efforts to focus on the most essential themes.
Response 3
As suggested, we reduced the text in the discussion to focus on the most essential themes (pages 17-21). Moreover, one paragraph of the former version of the manuscript (former #4.3) have been removed in the revised manuscript and one paragraph (former #4.4) has been incorporated into the last paragraph of the discussion (pages 20,21).
Point 4
Cdc42 is not only linked to cell polarity and the actin cytoskeleton, but also to Golgi trafficking and COPI coats (See papers by Mark Stamnes and Victor Hsu).
Response 4
As requested by the reviewer, we modified the text to mention that Cdc42 is linked to Golgi trafficking and COPI coats (lines 449-451, page 14) and cited the research articles by Mark Stamnes and Victor Hsu (see references #109-111).
Point 5
The data in Figure 7b are not quite clear. I'm not totally convinced of the preferential accumulation of GOLPH3 in the midzone (is the red signal specific?) Maybe staining for the microtubules of the intercellular bridge could give a better reference point? The PLA data in Figure 8 on the other hand appears more convincing, but does not add to the point regarding (co)localization of the proteins at the midzone.
Response 5
The accumulation of dGOLPH3 protein in the midzone of telophase dividing spermatocytes is specific, as demonstrated in our previous research article (Sechi et al.,2014, see ref #4). The accumulation of dGOLPH3 in the midzone is much more evident as the dividing cells progress from early telophase to mid-late telophase (see the cell at the bottom). The most appropriate fixing procedure to visualize dGOLPH3 at the cleavage furrow is reported in (Sechi et al.,2014, see ref #4) and requires formaldehyde and methanol. However, this procedure is not suitable for immunostaining with anti-dFmr1 antibodies.
To address the concern of the reviewer regarding (co)localization of the proteins at the midzone, we modified the text (Figure 7 legend and lines 518-520, page 16) to clarify that dFmr1 colocalizes with dGOLPH3-enriched organelles at the astral membranes of dividing spermatocytes.
Point 6 What is meant by "polar localization"? It seems to me that the GOLPH3 signal for instance is enriched at the sides of the nucleus facing the intercellular bridge.
Response 6
We clarified that dGOLPH3 is enriched in puncta at the astral membranes of dividing spermatocytes (Figure 7 legend and lines 518-520, page 16).
Point 7
Finally, one thing that I find very puzzling is that although GOLPH3 is commonly considered as a trans-Golgi/TGN protein (e.g. Sechi et al., 2020), it interacts with Rab1 (earlier data by the authors), ERGIC-53 and Sec22, all of which (at least in mammalian cells) are known to operate at the level of the ERGIC/cis-Golgi. Maybe the authors should point out this spatial dilemma, although future studies will certainly be required to find out the answer to this question.
Response 7
We addressed the concern of reviewer #1 on the spatial dilemma concerning the trans-Golgi/TGN localization of GOLPH3 and some molecular interactors located at the level of ERGIC/cis-Golgi. The discussion can be read on page 19 (lines 618-633).
Unavoidably, the description of the results (as well as the Discussion) in this type of paper becomes rather list-like. Maybe the authors should make more efforts to focus on the most essential themes.
Response 3
As suggested, we reduced the text in the discussion to focus on the most essential themes (pages 17-21). Moreover, one paragraph of the former version of the manuscript (former #4.3) have been removed in the revised manuscript and one paragraph (former #4.4) has been incorporated into the last paragraph of the discussion (pages 20,21).
Point 4
Cdc42 is not only linked to cell polarity and the actin cytoskeleton, but also to Golgi trafficking and COPI coats (See papers by Mark Stamnes and Victor Hsu).
Response 4
As requested by the reviewer, we modified the text to mention that Cdc42 is linked to Golgi trafficking and COPI coats (lines 449-451, page 14) and cited the research articles by Mark Stamnes and Victor Hsu (see references #109-111).
Point 5
The data in Figure 7b are not quite clear. I'm not totally convinced of the preferential accumulation of GOLPH3 in the midzone (is the red signal specific?) Maybe staining for the microtubules of the intercellular bridge could give a better reference point? The PLA data in Figure 8 on the other hand appears more convincing, but does not add to the point regarding (co)localization of the proteins at the midzone.
Response 5
The accumulation of dGOLPH3 protein in the midzone of telophase dividing spermatocytes is specific, as demonstrated in our previous research article (Sechi et al.,2014, see ref #4). The accumulation of dGOLPH3 in the midzone is much more evident as the dividing cells progress from early telophase to mid-late telophase (see the cell at the bottom). The most appropriate fixing procedure to visualize dGOLPH3 at the cleavage furrow is reported in (Sechi et al.,2014, see ref #4) and requires formaldehyde and methanol. However, this procedure is not suitable for immunostaining with anti-dFmr1 antibodies.
To address the concern of the reviewer regarding (co)localization of the proteins at the midzone, we modified the text (Figure 7 legend and lines 518-520, page 16) to clarify that dFmr1 colocalizes with dGOLPH3-enriched organelles at the astral membranes of dividing spermatocytes.
Point 6 What is meant by "polar localization"? It seems to me that the GOLPH3 signal for instance is enriched at the sides of the nucleus facing the intercellular bridge.
Response 6
We clarified that dGOLPH3 is enriched in puncta at the astral membranes of dividing spermatocytes (Figure 7 legend and lines 518-520, page 16).
Point 7
Finally, one thing that I find very puzzling is that although GOLPH3 is commonly considered as a trans-Golgi/TGN protein (e.g. Sechi et al., 2020), it interacts with Rab1 (earlier data by the authors), ERGIC-53 and Sec22, all of which (at least in mammalian cells) are known to operate at the level of the ERGIC/cis-Golgi. Maybe the authors should point out this spatial dilemma, although future studies will certainly be required to find out the answer to this question.
Response 7
We addressed the concern of reviewer #1 on the spatial dilemma concerning the trans-Golgi/TGN localization of GOLPH3 and some molecular interactors located at the level of ERGIC/cis-Golgi. The discussion can be read on page 19 (lines 618-633).
Reviewer 2 Report
This is a competent piece of work to examine the interactome of Golgi phosphoprotein 3 (GOLPH3), a conserved protein that binds to the Golgi membrane in a manner dependent upon phosphatidylinositol 4-phosphate. The paper has archival value through identifying proteins that associate with GOLPH3. These include proteins involved in vesicle-mediated trafficking, cell proliferation and cilium formation. It is a vslusble study that should be well cited and is suitable for publication in Cells without necessitating revision.Author Response
Point #1
This is a competent piece of work to examine the interactome of Golgi phosphoprotein 3 (GOLPH3), a conserved protein that binds to the Golgi membrane in a manner dependent upon phosphatidylinositol 4-phosphate. The paper has archival value through identifying proteins that associate with GOLPH3. These include proteins involved in vesicle-mediated trafficking, cell proliferation and cilium formation. It is a vslusble study that should be well cited and is suitable for publication in Cells without necessitating revision.
Response #1
We thank the reviewer for appreciating our work.
Reviewer 3 Report
The Giansanti group has presented a comprehensive analysis of the GOLPH3 interactome in Drosophila testes that provides complementary evidence in support of previous studies on this interesting Golgi glycoprotein. In a somewhat surprising twist, this detailed analysis of the interactome suggests dGOLPH3 interacts with the Drosophila orthologs of Fragile X mental 25 retardation protein and Ataxin-2, suggesting a potential role in the pathophysiology of disorders of 26 the nervous system. Thus, the title focuses on the latter but since the detailed interactome indicates so many binding partners (whether indirect or direct) it seems that the title is a bit of a stretch.
Technically, this report is very well done and carefully controlled. The data is presented well and well documented. The authors are to be complemented on the exhaustive analysis that had to be a huge undertaking. The comprehensive nature of the analysis should be of great interest to both the membrane trafficking field as well as the cytoskeleton field. Again, much of the developed interactome is confirmatory of known interactions, which, in many ways, validates the methods. In terms of the writing, it may be worth adding some more of the primary literature on some of the cell biological pathways. For instance, much has been written about the recycling endosomal pathway in cytokinesis in flies, worms and sea urchins. The same can be said about the coupling of dynein to membrane organelles. While the interactome is extensive and demonstrates partners across several pathways, one suggestion is to be conservative as to the central role of dGOLPH3 as the central player in regulating these pathways.
Author Response
Point #1
The Giansanti group has presented a comprehensive analysis of the GOLPH3 interactome in Drosophila testes that provides complementary evidence in support of previous studies on this interesting Golgi glycoprotein. In a somewhat surprising twist, this detailed analysis of the interactome suggests dGOLPH3 interacts with the Drosophila orthologs of Fragile X mental 25 retardation protein and Ataxin-2, suggesting a potential role in the pathophysiology of disorders of 26 the nervous system. Thus, the title focuses on the latter but since the detailed interactome indicates so many binding partners (whether indirect or direct) it seems that the title is a bit of a stretch.
Response 1
We thank the reviewer for appreciating our work. Although the detailed interactome indicates many binding partners we think that dGOLPH3 interaction with dFmr1 and Ataxin-2, well supported by Co-IP, GST pull-down and PLA experiments, is intriguing and can open a new field of investigation correlated with GOLPH3 function in neurological diseases. Moreover, since human FMRP has been involved in different cancer types, it will be important to investigate the functional dependence between GOLPH3 and FMRP in the light of a therapeutic strategy of human cancer. For this reason, we think that the title is appropriate
Point #2
In terms of the writing, it may be worth adding some more of the primary literature on some of the cell biological pathways. For instance, much has been written about the recycling endosomal pathway in cytokinesis in flies, worms and sea urchins. The same can be said about the coupling of dynein to membrane organelles. While the interactome is extensive and demonstrates partners across several pathways, one suggestion is to be conservative as to the central role of dGOLPH3 as the central player in regulating these pathways.
Response # 2
As suggested by reviewer #2, we modified the discussion and added some more primary literature on the role of endocytic and recycling endosomal pathways in cytokinesis in model organisms and mammalian cells (lines 566-592, page 18). In addition, we reduced the text in the discussion to focus on the most essential themes such as the central role of GOLPH3 in vesicle trafficking during cytokinesis (pages 17-21).